# Clinical and Radiological Outcomes of Accessory Renal Artery Exclusion during Endovascular Repair of Abdominal Aortic Aneurysms

**DOI:** 10.3390/diagnostics14090864

**Published:** 2024-04-23

**Authors:** Alessia Di Girolamo, Marta Ascione, Francesca Miceli, Alireza Mohseni, Chiara Pranteda, Pasqualino Sirignano, Maurizio Taurino, Luca di Marzo, Wassim Mansour

**Affiliations:** 1Vascular and Endovascular Surgery Division, Department of General Surgery and Surgical Specialties, Policlinico Umberto I, Sapienza University of Rome, Viale del Policlinico, 155, 00161 Rome, Italy; alessia.digirolamo@hotmail.it (A.D.G.); marta.ascione@uniroma1.it (M.A.); francesca.miceli@uniroma1.it (F.M.); luca.dimarzo@uniroma1.it (L.d.M.); 2Faculty of Medicine and Surgery, Sapienza University of Rome, Viale Regina Elena, 324, 00161 Rome, Italy; mohseni.1974637@studenti.uniroma1.it; 3Vascular and Endovascular Surgery Unit, Sant’Andrea Hospital, Sapienza University of Rome, Via di Grottarossa 1035/1039, 00189 Rome, Italy; chiara.pranteda@uniroma1.it (C.P.); pasqualino.sirignano@uniroma1.it (P.S.); maurizio.taurino@uniroma1.it (M.T.)

**Keywords:** abdominal aortic aneurysm, endovascular aneurysm repair, accessory renal artery, embolization, renal function

## Abstract

Background: Accessory renal arteries (ARAs) frequently coexist with abdominal aortic aneurysms (AAA) and can influence treatment. This study aimed to retrospectively analyze the ARA’s exclusion effect on patients undergoing standard endovascular aneurysm repair for AAA. Methods: The study focused on medium- and long-term outcomes, including type II endoleak, aneurysmal sac changes, mortality, reoperation rates, renal function, and infarction post-operatively. Results: 76 patients treated with EVAR for AAA were included. One hundred and two ARAs were identified: 69 originated from the neck, 30 from the sac, and 3 from the iliac arteries. The ARA treatment was embolization in 15 patients and coverage in 72. Technical success was 100%. One-month post-operative computed tomography angiography (CTA) revealed that 76 ARAs (74.51%) were excluded. Thirty-day complications included renal deterioration in 7 patients (9.21%) and a blood pressure increase in 15 (19.73%). During follow-up, 16 patients (21.05%) died, with three aneurysm-related deaths (3.94%). ARA-related type II endoleak (T2EL) was significantly associated with the ARA’s origin in the aneurysmatic sac. Despite reinterventions were not significantly linked to any factor, post-operative renal infarction was correlated with an ARA diameter greater than 3 mm and ARA embolization. Conclusion: ARAs can influence EVAR outcomes, with anatomical and procedural factors associated with T2EL and renal infarction. Further studies are needed to optimize the management of ARAs during EVAR.

## 1. Introduction

Accessory renal arteries (ARAs) are frequently observed in conjunction with abdominal aortic aneurysms (AAA). Among patients undergoing endovascular aneurysm repair (EVAR), the presence of ARAs is noted in approximately 9.5% to 16.2%, translating to an estimated prevalence of 12% to 25% [1]. These ARAs can originate from the aneurysmal neck, sac, or, less commonly, the iliac arteries. However, it is pivotal to note that only those ARAs that arise distally from the primary renal artery hold clinical significance for endovascular repair, as they can influence the treatment choice.

The coverage or embolization of ARAs might be essential to ensure the integrity of the endoprosthesis seal and mitigate the risk of endoleak after EVAR. Such interventions, while crucial, carry the potential risk of post-operative renal function deterioration [2]. This is further compounded by the inherent association of EVAR with renal dysfunction, attributed to the use of iodine contrast medium (iCM) and the intricacies of intraluminal manipulations [3].

Current guidelines from the European Society for Vascular Surgery (ESVS) advocate preserving ARAs larger than 3 mm or supplying more than one-third of the renal parenchyma, while the Italian Society for Vascular Surgery (SICVE) guidelines suggest accessory renal coverage to achieve an infrarenal neck adequate for EVAR, enhancing the complicated scenario in which our study was performed [4].

In fact, prevailing evidence leans towards covering the vessel, especially when it is situated at the neck level, to achieve an immediate and robust sealing zone, enhancing sac stability. Nevertheless, accessory renal artery coverage encounters some risks, and to prevent them, several strategies exist for ARA preservation, including reimplantation during open surgery, using custom-made or fenestrated devices (FEVAR) [5], and revascularization employing parallel graft techniques [6]. Notably, the reimplantation of ARAs is primarily suggested to safeguard renal function, especially in patients diagnosed with chronic renal failure (CKD). 

The aim of the present study was to evaluate the impact of ARA exclusion in patients submitted to standard EVAR in a retrospective series. 

The emphasis is on discerning medium- to long-term outcomes, encompassing aspects such as type II endoleak (T2EL), alterations in the aneurysmal sac, mortality rates, reoperation frequencies, instances of renal infarction, and immediate post-operative and extended renal function trajectories.

## 2. Materials and Methods

This study encompassed patients who underwent elective EVAR for AAA and concurrently exhibited an ARA originating from the neck, aneurysmal sac, or iliac arteries. The research from March 2015 to March 2022 was conducted at two academic institutions: Sapienza’s University Hospitals, specifically the Policlinico Umberto I and Sant’Andrea Hospitals in Rome.

All patients submitted to EVAR in an elective setting between March 2015 and March 2022, who presented an Accessory Renal Artery (ARA) at the preoperative CTA, were included in the present study. 

ARA was defined as additional arteries that pass along with normal renal arteries through the hilum, are smaller than the principal vessel by 3 mm or more, and supply no more than 1/3 of the renal parenchyma. 

Patients requiring urgent or emergency interventions for ruptured or symptomatic AAA, those with juxtarenal, pararenal, or thoracoabdominal aneurysms, graft infections, or a history of aortic surgeries, were excluded from the study. Data collected retrospectively underwent prospective analysis. The study assessed preoperative risk factors, including arterial hypertension, diabetes, coronary artery disease (CAD), dyslipidemia, chronic kidney disease (CKD), and smoking habits. Renal function was ascertained using serum creatinine levels and creatinine clearance during preoperative and post-operative phases. The established normal range was 0.5–1.2 mg/dL for serum creatinine and <60 mL/min for creatinine clearance, as determined by the Cockroft-Gault formula. Patients diagnosed with CKD were categorized according to the 2012 Kidney Disease Improving Global Outcome (KDIGO) Guidelines. Patients presenting with serum creatinine levels exceeding 1.2 mg/dL were administered an intravenous saline infusion at 1 mg/kg/h for 12 h preoperatively and 12 h post-operatively. 

For elevated creatinine levels surpassing 1.5 mg/dL, or GFR < 45 mL/min/1.73 mq, the nephrological protocol indicated intravenous saline infusion at 1 mL/kg/h for 12 h preoperatively and 12 h post-operatively, or NaCO_3_ 1.4% at 3 mL/kg/h 1 h before intraarterial contrast medium administration and 1 mL/kg/h during the procedure and for the next 6 h. 

All preoperative CTAs were assessed using the OsiriX MD software 12.5.2, using multiplanar reconstruction. This assessment aimed to pinpoint ARAs, discern their origin, and aid in preoperative planning. All computed tomography was contrast-enhanced; however, not all the CT scans were 1 mm sliced. 

The endoprosthesis selection was predicated upon the aortic aneurysm’s morphology, while the vessel’s origin and diameter determined the treatment modality for ARAs. 

The embolization procedure was discussed by the vascular surgeon team. Anesthesia modalities encompassed local, spinal, or general options. Each procedure was executed in a specialized operating room equipped with a mobile C-arm. All procedures were performed by vascular surgeons. The volume of contrast medium utilized during the endovascular procedure was documented. Technical success was delineated by the successful implantation of a stent graft without the need for surgical conversion, intraoperative mortality, type I or III endoleaks, or any evidence of stent graft migration or occlusion right after the operation.

In the post-operative phase, monitoring was employed to detect acute kidney injury (AKI), defined by an elevation in serum creatinine levels by 0.3 mg/dL within the initial 48 h post-intervention, significant blood pressure deviations, fever, and lumbar or flank pain. The duration of the post-operative intensive care unit (ICU) stay and the overall hospitalization period were carefully recorded. Subsequent CTAs were analyzed to confirm the accurate exclusion of ARAs, detect renal infarction, ascertain aneurysm diameter, and identify the presence and type of endoleak.

The follow-up protocol mandated a CTA 1-month post-operation and at the 1-year mark, supplemented by routine blood tests, including serum creatinine levels. DUS was performed at six- and 12-month intervals and subsequently annually. Ambiguous or non-diagnostic DUS results warranted a CTA. Contrast-enhanced ultrasound (CEUS) is not performed routinely in our center. 

Patients underwent systematic evaluations for reinterventions, renal function, and mortality during these follow-up sessions. Imaging assessments focused on the aneurysmal sac’s diameter and its evolution.

### 2.1. Study Outcomes 

The impact of ARA exclusion was considered in terms of type II endoleak development, reintervention rate, aneurysmal sac evolution, mortality, renal infarction, acute post-operative kidney injury, and long-term chronic kidney disease. 

Endpoints could be divided into two groups: The aneurysm-related endpoints and the renal-related endpoints. Type II endoleak was researched on post-operative and subsequent CTA and correlated with reintervention, aneurysmal sac enlargement, and AAA mortality. On the other hand, renal infarction was detected both clinically and on post-operative CTA and was correlated with immediate post-operative serum creatinine worsening and long-term chronic kidney disease.

### 2.2. Statistical Analysis 

Comparative analyses were executed using the χ2 test and Fisher’s exact test, contingent upon the data. IBM SPSS Statistics for Windows, Version 25, was the tool of choice for statistical analysis. Continuous variables were articulated as means +/− standard deviation, while categorical variables were represented as percentages. A *p* value of ≤0.05 was the threshold for statistical significance.

## 3. Results

During the reported period, 1252 patients were treated by EVAR. Among them, 76 patients treated with EVAR for infrarenal AAA were included in this study. Seventy were male (92%) and six were female (8%). At the time of intervention, the mean age was 72.75 years +/− 7.61 (range 46–86). Risk factors are summarized in Table 1. 

Preoperative serum creatinine and clearance were respectively 1.11 mg/dL +/− 0.37 and 72.74 mL/min +/− 23.8. Analyzing the preoperative renal function, 24 patients presented with CKD: 10 patients (41.3%) were stage I, 13 patients (54.2%) stage II, and only 1 (4.2%) stage III. 

At the preoperative CTA analysis, the mean diameter sac was 51.6 mm +/− 12.82. One hundred and two ARAs were identified: 69 (67.64%) originated from the aortic neck, 30 (29.41%) from the aneurysmal sac, and 3 (2.94%) from the common iliac arteries (Table 2). 

The anesthesia performed was local in 21 patients, spinal in 2 patients, and general in 53 patients. Mean contrast medium used was 70.22 cc +/− 28.86. The chosen endoprosthesis is summarized in Table 3. The only patient submitted to EVAR with the Nellix System was excluded from this analysis because the device was not comparable to other EVARs and had known worse long-term results [7,8].

All endoprostheses, chosen according to the anatomy as underlined in Section 2, were implanted inside the IFU. 

The ARA treatment, performed during the EVAR procedure, was embolization in 15 patients, using metallic coils in 11 cases, an Amplatzer plug in 4, and coverage in 72. Technical success was 100%. 

Analyzing post-operative CTA, 76 ARAs (75.24%) were excluded, while 25 (24.75%) were left untreated. 

### 3.1. In-Hospital Results

During the immediate post-operative period, no deaths were registered. Renal function worsening was experienced in 7 patients (9.21%). Of these patients, 5 (71.43%) were preoperatively affected by CKD, and only one patient was submitted to ARA embolization, while the others were submitted to coverage. No dialytic treatment was required. In 15 patients (19.73%), a blood pressure increase was reported, requiring an antihypertensive treatment change in 13. During the post-operative in-hospital stay, 2 patients developed fever (2.63%) and 3 patients referred back pain, which resolved spontaneously after 24 h. Twenty-two patients (28.94%) stayed in the ICU post-operatively for 24 h, while the mean length of hospital stay was 8.78 days +/− 7.1. No reinterventions were needed.

### 3.2. 30-Day Outcomes

No deaths were recorded. On 30-day post-operative CTA, no type I or III endoleak was detected; T2EL was noted in 18 patients (23.68%), and renal infarction with ARA anatomical correspondence distribution was observed in 18 patients (23.68%). Among patients who presented with renal infarction, antihypertensive treatment change was needed in 4 patients (22.2%). ARA involvement detected on CTA was present in 6 patients (33.3%) presenting T2EL, and other branches were concomitantly involved in all cases. In 2 patients, 3 couples of lumbar arteries were involved, and in the remaining 4, 2 couples of lumbar arteries and an IMA were involved. An aneurysmal sac enlargement greater than 5 mm was present in 1 patient. In the subsequent follow-up controls, the aneurysmal sac enlarged in 2 patients, needing further interventions. No reinterventions were needed. Serum creatinine worsening was still present in 5 patients, preoperatively affected by CKD, and only one patient was submitted to ARA embolization, while the others were submitted to coverage. No dialytic treatment was required. No change in blood pressure or antihypertensive treatment was detected.

### 3.3. Mean Follow-Up

Median follow-up was 23.81 months (range: 1–108 months). During follow-up, 16 patients died (21.05%), and aneurysm-related death was recorded in 3 cases (3.94%). Reinterventions were reported in 3 patients. One patient, presenting T2EL and sac enlargement of more than 1 cm, with ARA, IMA, and lumbar artery involvement, was firstly submitted to sac and ARA embolization with metallic coils and glue. After this, the persistence of T2EL leads to a second procedure of saccotomy with lumbar arteries, IMA, and ARA ligation. Nevertheless, the persistence of T2EL and sac enlargement led to neck evolution with the development of type IA endoleak, treated with a proximal aortic cuff placement with bilateral renal stenting using the Chimney technique. Another patient, presenting with T2EL, involving both ARA and lumbar arteries, and sac enlargement, was subjected to lumbar artery embolization with metallic coils. The following CTA controls showed sac stability. In 2 patients, a distal relining with an iliac extension placement for type Ib endoleak was required.

During follow-up, 9 patients out of 76 were lost to follow-up visits, with a significant drop-out rate (11.8%). 

Out of 67 patients evaluated during the follow-up period, 41 (61.2%) presented sac stability, 23 (34.3%) presented sac regression, and only 3 (4.5%) presented sac instability. At the last follow-up, 5 patients (6.57%) presented a creatinine serum worsening of more than 0.3 mg/dL. 

Patients with an ARA greater than 3 mm were more likely to receive an embolization treatment (OR 16.50, 95% CI 2.040–133.444, *p* < 0.001), and a diameter greater than 3 mm was as well related to renal infarction (OR 5.74, 95% CI 1.499–21.983, *p* = 0.006) and to T2EL development (OR 5.74, 95% CI 1.499–21.983, *p* = 0.006).

Renal infarction was associated with ARA embolization (OR 3.98, 95% CI 1.190–13.288, *p* = 0.019) and post-operative renal function worsening (OR 3.83, 95% CI 1.249–11.764, *p* = 0.015), but no statistically significant associations were observed with ARA coverage, ARAs number, development of hypertension, or changes in antihypertensive therapy post-operatively, and it was not correlated with long-term serum creatinine worsening nor with long-term mortality. 

Post-operative AKI was however associated with the presence of preoperative CKD (OR 5.90, 95% CI 1.249–11.764, *p* = 0.015), as well as with ARA diameter greater than 3 mm and exclusion, but analyzing long-term post-operative serum creatinine worsening, no associations emerged with ARA exclusion, ARA diameter greater than 3 mm, or the amount of contrast medium used. 

T2EL was associated with an ARA diameter greater than 3 mm (OR 5.74, 95% CI 1.499–21.983, *p* = 0.006) and an ARA origin from the aneurysmal sac (OR 3.14, 95% CI 1.044–9.465, *p* = 0.037). 

In our cohort, only 6 reinterventions occurred in 3 patients, and four reinterventions were T2EL related. However, small numbers do not lead to any significant association, except for overall mortality and AAA-related mortality (OR 8.43, 95% CI 0.713–99.662, *p* = 0.048, and OR 17.75, 95% CI 1.099–286.557, *p* = 0.008, respectively). 

Mortality was not associated with ARA exclusion (OR 0.9, 95% CI 0.09–9.69, *p* = 0.95), with post-operative AKI (OR 1.31, 95% CI 0.22–7.53, *p* = 0.75), hospital length of stay (OR 2.99, 95% CI 0.92–9,0, *p* = 0.06), post-operative intensive care (OR 1.25, 95% CI 0.37–4.16, *p* = 0.71), renal infarction (OR 1.64, 95% CI 0.43–5.58, *p* = 0.42), type II endoleak (OR 2, 95% CI 0.72–7.91, *p* = 0.14), and sac instability (OR 7.14, 95% CI 0.6–84.66, *p* = 0.075).

All statistical findings are summarized in Table 4.

## 4. Discussion

Endovascular aneurysm repair (EVAR) has emerged as the therapeutic approach for infrarenal abdominal aortic aneurysms (AAAs) due to its minimally invasive nature compared to traditional open surgery. Nonetheless, reintervention rates associated with EVAR, documented as 7–19%, present a significant limitation [9,10]. The primary cause for these reinterventions has been identified as type II endoleak, associated with complications such as long-term sac expansion, additional interventions, potential delayed rupture, and surgical conversion.

Accessory renal arteries (ARAs) are frequently identified during preoperative evaluations, with incidences reported between 9.5% and 16.2% [1]. The debate regarding the necessity of ARA coverage persists, with the feasibility of such coverage often contingent upon the anatomical location of the ARA.

A T2EL rate surpassing 20% was noted in the cohort under investigation. Remarkably, one-third of these endoleaks were associated with ARA involvement, often in tandem with the inferior mesenteric artery and at least one pair of lumbar arteries, emphasizing the crucial role of efferent branches in sac reperfusion. 

Recent literature has indicated that a substantial proportion of ARAs, approximately 70%, originate from the proximal neck. In contrast, 29% are found to arise from the aneurysm sac, with a mere 3% from the iliac arteries, a sporadic occurrence in clinical observations [11,12]. The significance of other branches, both afferent and efferent to the sac, for T2EL development has been highlighted in various studies, and vessel diameter has been pinpointed as a critical risk factor for developing endoleaks and subsequent complications [13,14,15].

Piazza et al. summarized the importance of both diameter and number of branches in a classification of patients at risk of post-operative T2EL development: when a patent IMA with a diameter of more than 3 mm was present, 3 pairs of lumbar arteries were patent, or 2 lumbar arteries were patent and associated with a sacral artery, an accessory renal artery, and/or any diameter IMA, or when any of the above criteria plus any patent aortic branch were present [16].

In fact, complex interactions among multiple collateral vessels can resemble arteriovenous malformations, with inflow and outflow branches determined by the pressure gradient between the aorta and each branch [17,18,19]. 

The statistical analysis demonstrated a that there is a significant risk for the development of type II endoleaks when the ARA emanates from the aneurysm sac. This association was further accentuated when focusing solely on endoleaks directly linked to ARAs. Several studies have explored the implications of ARA coverage. It was observed that coverage at the aortic neck level might predispose to vessel thrombosis. However, such outcomes were less probable when the ARA was found to emerge from the aneurysm sac [20,21,22,23]. Malgor et al. highlighted that nearly one-third of EVAR procedures led to type II endoleaks, which became persistent when a patent ARA exceeding 3 mm in diameter emerged from the aneurysm sac [24]. Such endoleaks were associated with sac expansion, leading to the need for reintervention. In contrast, when ARAs were found to emerge from the aneurysm neck, endoleaks were not reported. Given these observations, recommendations were made for intraoperative embolization of ARAs larger than 3 mm that originated from the aneurysm sac. However, such interventions were deemed unnecessary for patients with ARAs emerging from the neck and subsequently covered by the stent graft [25].

Data from the cohort under study supported these findings, suggesting a significant association between an ARA diameter exceeding 3 mm and the onset of type II endoleaks. Rokosh et al. provided evidence indicating that patients undergoing EVAR in conjunction with efferent vessel embolization experienced more frequent long-term sac regression than those undergoing aneurysm exclusion alone. Yet, this did not correspond to a decreased incidence of endoleaks [26]. O’Donnell et al. proposed that failure of sac regression post-EVAR might be more prevalent in patients undergoing the procedure outside the Instructions for Use (IFU) [27]. In the cohort examined, even though the majority had ARAs originating from the neck, the sole type IA endoleak was most likely a secondary outcome of a type II endoleak.

From a renal perspective, the potential consequences of ARA coverage have been extensively examined, especially concerning the risk of renal infarction and subsequent renal function deterioration. The literature presents a diverse range of outcomes in this regard. Sadeghi-Azandaryani et al. observed that there was a significant eGFR decrease after 1 week and 6 months in patients with ARA when compared with patients without ARA. However, this decrease is not significant for later follow-up control. The authors also found out that renal function worsening was present in patients with preoperatively normal renal function when compared to previously impaired renal function. 

On one hand, these study results are in line with the literature results expressed above; in fact, post-operative AKI was associated with ARA embolization but did not affect longer-term post-operative renal function. On the contrary, in our cohort, post-operative AKI was associated with preoperative CKD. 

Renal infarction was significantly associated with ARAs larger than 3 mm in diameter. Aquino et al. postulated that the low incidence of renal infarction might be due to judicious selection criteria, wherein only ARAs of smaller caliber were considered for exclusion.

In complex thoraco-abdominal endovascular repair with fenestrated custom-made devices, it has also been demonstrated that ARA incorporation is feasible, with low complications and good primary assisted patency. However, in case of infrarenal AAA, the construction of a custom-made fenestrated graft, despite feasible, will lead to increased complexity and device costs. Nevertheless, no adequate bridging stent are available on the market for small accessory renal artery [28].

Our study has some obvious limitations: First of all, data collection was obtained in a retrospective design and can therefore be influenced by differences in daily clinical practice, despite being treated according to standard protocols. Secondly, the sample size is small but comparable with previous studies. Another important limitation is the absence of a control group without ARA to compare results both in terms of renal function and T2EL. 

Lastly, the inherent limitations of computed tomography (CT) scans must be acknowledged, especially concerning accurately measuring smaller vessel diameters, such as ARAs. Despite meticulous image review by multiple examiners, potential inaccuracies due to resolution constraints cannot be overlooked; in fact, all CT scans were contrast-enhanced, but as described in the Method section, not all of them were sliced at 1 mm.

While the implications of ARA coverage on renal function and other outcomes remain subjects of ongoing research and debate, this study offers valuable insights that can inform clinical decision making and future research directions.

In fact, the aim of post-operative imaging is to predict or detect complications, and various imaging modalities can be used during EVAR follow-up. As enhanced by our latest guidelines, some anatomical factors have been found to predict later complications, and the follow-up timing should be differentiated based on the first 30-day post-operative CTA. Patients presenting ARA on preoperative CTA should undergo semestral DUS and annual CTA for the first 5 years, because ARAs presence should be seen as a preoperative risk factor for T2EL development. 

## 5. Conclusions

The efficacy of EVAR in patients possessing an ARA seems to be influenced by certain anatomical features. Notably, the origin of the aneurysm sac and a diameter exceeding 3 mm have been identified as factors correlating with an increased risk of type II endoleak. Given these findings, it is suggested that intraoperative embolization of the ARA be considered for patients exhibiting the aforementioned morphological characteristics. Such an intervention may aid in preventing endoleaks and subsequent reinterventions, fostering enhanced sac stability over extended periods. Moreover, a heightened post-EVAR monitoring regimen is recommended for this patient subset. 

Exclusion of the ARA was not observed to lead to prolonged renal function deterioration post-EVAR in the analyzed cohort. However, notable perioperative deterioration was documented in individuals presenting with pre-existing chronic renal failure.

It is imperative to conduct additional research encompassing a more expansive patient cohort to ascertain these patients’ post-operative outcomes and determine the optimal surgical strategy. Such studies should ideally compare outcomes following ARA exclusion to those where the ARA is preserved during the EVAR procedure.

## Figures and Tables

**Table 1 diagnostics-14-00864-t001:** Cardiovascular risk factors.

Risk Factors	N°
Male	70/76 (92%)
Female	6/76 (8%)
Arterial hypertension	50/76 (65%)
Diabetes	20/76 (26.31%)
CAD	19/76 (25%)
Dyslipidemia	45/76 (59.21%)
Cronic kidney disease	24/76 (31.57%)
Smoke babit	36/76 (47.36%)

**Table 2 diagnostics-14-00864-t002:** Accessory renal artery characteristics.

Total ARA	102
ARA from the neck	69/102 (67.64%)
ARA from the aneurysmal sac	30/102 (29.41%)
ARA from iliac arteries	3/102 (2.94%)
1 ARA	57/76 (75%)
2 ARAs	14/76 (18.42%)
3 or more ARAs	5/76 (6.58%)
Mean ARA diameter (mm)	1.87
ARAs greater than 3 mm	46/102 (45.09%)

**Table 3 diagnostics-14-00864-t003:** Endoprosthesis used during EVAR procedure.

Graft Type	N°
Gore Excluder	34
Medtronic Endurant	12
Cook Alpha	10
Endologix AFX	13
AFX cuff	1
Endurant cuff	2
Ovation Alto	5
Nellix	1
Cordis Incraft	1

**Table 4 diagnostics-14-00864-t004:** Statistical findings.

		OR	95% CI	*p*-Value
ARA > than 3 mm	Embolization treatment	16.50	2.040–133.444	<0.001
	Renal infarction	5.74	1.499–21.983	0.006
	T2EL development	5.74	1.499–21.983	0.006
Renal infarction	ARA embolization	3.98	1.190–13.288	0.019
	Post-operative renal function worsening	3.83	1.249–11.764	0.015
Post-op AKI	Preoperative CKD	5.90	1.249–11.764	0.015
T2EL	ARA diameter greater than 3 mm	5.74	1.499–21.983	0.006
	ARA origin from the aneurysmal sac	3.14	1.044–9.465	0.037
	ARA origin from the neck	1.46	0.49–4.31	0.48
	ARA origin from the iliac arteries	1.64	0.14–19.29	0.68
	ARA coverage	1.6	0.17–14	0.67
	ARA exclusion	1.64	0.14–19.29	0.69
	ARA embolization	1.37	0.37–5.05	0.63
Reintervention	Overall mortality	8.43	0.713–99.662	0.048
	AAA-related mortality	17.75	1.099–286.557	0.008
Mortality	ARA exclusion	0.9	0.09–9.69	0.95
	Post-operative AKI	1.31	0.22–7.53	0.75
	Hospital length of stay	2.99	0.92–9.0	0.06
	Post-operative ICU	1.25	0.37–4.16	0.71
	Renal infarction	1.64	0.43–5.58	0.42
	Type II endoleak	2	0.72–7.91	0.14
	Sac instability	7.14	0.6–84.66	0.075

## Data Availability

Data are not available due to privacy restrictions.

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
