# Peer review of "Clinical and Radiological Outcomes of Accessory Renal Artery Exclusion during Endovascular Repair of Abdominal Aortic Aneurysms"

_diagnostics, 2024, doi:10.3390/diagnostics14090864_

Round 1

Reviewer 1 Report

Comments and Suggestions for Authors

Dear authors,

congratulations on this interesting manuscript on the relevance and outcomes of ARAs during EVAR. I have however several questions and comments that can be found below:

Methods: Who performed the EVAR and embolization procedures – I assume vascular surgeons? Or radiologist or together?

Who decided about the procedure regarding ARAs? Purely the operating surgeon or was this discussed within the vascular surgical or a multidisciplinary team?

Did the ultrasound follow-up also include contrast-enhanced ultrasound (CEUS), which is known to have a higher sensitivity to detect slowly flowing type II endoleaks compared to CTA ? Were the kidneys also assessed specifically during the ultrasound follow-up regarding perfusion and infarction, maybe also with CEUS? If not, these two aspects are worth adding to the discussion.

Results:

Table 1: Table 1 has to be named more precisely, e.g. cardiovascular risk factors. What is meant by smoke habit? Current smoker or history of smoking or both? Are the pack years known?

I assume that ARAs arising from the iliac arteries were all arising from the common iliac arteries? Please specify.

Were all the EVARs implanted inside IFU? The Nellix System is not comparable to other EVARs and also with respect to long-term results and endoleaks it should not be included in this analysis.

In-Hospital results: What was your perioperative kidney protection regime in patients with known CKD undergoing EVAR? Please add this to the methods section. Did you perform no imaging control (CTA and/or ultrasound) in the patients before discharge?

Lines 178-181: Was ARA treatment performed simultaneously with EVAR or in a separate procedure? A technical success of 100% seems contradictory with 25% of ARA in postoperative CTA scan resulting untreated. Please clarify this part. If 25% of ARA result still patent despite occlusion treatment this is highly relevant and has to be discussed further including comparison to existing literature and also mentioned in the conclusion.

Line 206: Follow-up should be reported either as mean +- standard deviation or as median and range. Further, with a mean FU of about 24 months you can only speak of mid-term but not long-term outcomes. Please also add information about completeness of FU / patient drop-outs.

Line 208-2014: You describe first ARA and lumbar artery involvement, then later also the IMA was also ligated. Was the IMA also contributing to the type 2 EL from the beginning? Or was there no retrograde perfusion from IMA to the aneurysm sac in the beginning? Often, in such cases with aneurysm sac growth despite type 2 EL treatment, unrecognized type Ia or Ib endoleak was present from the beginning. In such cases, additional CEUS may help in the diagnostic process. What type of EVAR was implanted in this case, was it inside IFU regarding proximal neck characteristics?

Line 217: Why was distal extension required in two patients? Type Ib endoleaks?

In general: How many of your patients with type 2EL had communications between ARA and/or lumbar arteries and/or the IMA? And have you got information on anticoagulation therapy in the patients of your study cohort? Both aspects are important to note and discuss, since they may contribute to endoleak persistence requiring embolization. I suggest adding this to your analysis.

Discussion:

Lines 268-274: Here you discuss correctly the relevance of communicating arteries in type 2 EL persistence with respective references. However, the prevalence of multiple arteries in your study should be described more precisely, in the methods section (see commentaries above) and here.

Line 297-299: I suggest rephrasing this sentence as “…was most likely a secondary outcome…” because you cannot be entirely sure that this was the case. Even if CT scans did not show contrast media in the aortic neck at the level of the ARA, may still have contributed to the type IA endoleak in terms of endotension.

Lines 300-305: Please discuss your results compared to those of Sadeghi-Azandaryani et al.

Lines 309-312: How was the resolution of CT scans in your series? Were they all 1mm slices and were they all contrast-enhanced? This has to be clarified also in the methods section.

Line 305: “2” has to be removed

Line 309: “17” has to be removed

You should comment on kidney-protection perioperative strategies to prevent renal function deterioration in case of ARA covering/coilig. Please also comment the possibility of ARA preservation by FEVAR procedures versus the risk of occlusion of small stented ARAs in FEVAR.

Further, since you recommend in the conclusion a heightened post-EVAR monitoring regimen, you should expand on this in the discussion. How would it look like?

Discussion of study limitations is largely missing, please revise.

Comments on the Quality of English Language

The manuscript is well written and requires only a minor spell- and grammar check.  

Author Response

Dear reviewer, 

We have revised the Manuscript according to your suggestions and we have answered each comment below.

Methods: Who performed the EVAR and embolization procedures – I assume vascular surgeons? Or radiologist or together? All procedures were performed by vascular surgeons

Who decided about the procedure regarding ARAs? Purely the operating surgeon or was this discussed within the vascular surgical or a multidisciplinary team? The embolization procedure was discussed by the vascular surgeon team.

Did the ultrasound follow-up also include contrast-enhanced ultrasound (CEUS), which is known to have a higher sensitivity to detect slowly flowing type II endoleaks compared to CTA ? Were the kidneys also assessed specifically during the ultrasound follow-up regarding perfusion and infarction, maybe also with CEUS? If not, these two aspects are worth adding to the discussion. Contrast-enhanced ultrasound (CEUS) is not performed routinely in our center.

Results: 

Table 1: Table 1 has to be named more precisely, e.g. cardiovascular risk factors. The table  name has been modified. What is meant by smoke habit? Current smoker or history of smoking or both? Are the pack years known? For smoke habit, it was meant both the current smokers and the history of smoking. Unfortunately the pack/years are not known.

I assume that ARAs arising from the iliac arteries were all arising from the common iliac arteries? Please specify. It has been specified.

Were all the EVARs implanted inside IFU? This topic has been highlighted.

The Nellix System is not comparable to other EVARs and also with respect to long-term results and endoleaks it should not be included in this analysis. The patient submitted to EVAR with the Nellix System has been removed from the analysis and it has been written in the Result section. However the patient in-hospital stay was uneventful and the patient is one out of 9 that was lost to follow-up. The data concerning the analysis has been revised accordingly.

In-Hospital results: What was your perioperative kidney protection regime in patients with known CKD undergoing EVAR? Please add this to the methods section. It has been added.

Did you perform no imaging control (CTA and/or ultrasound) in the patients before discharge? No, patients are submitted to 1 month-CTA

Lines 178-181: Was ARA treatment performed simultaneously with EVAR or in a separate procedure? ARAs treatment was performed during EVAR. A technical success of 100% seems contradictory with 25% of ARA in postoperative CTA scan resulting untreated. Please clarify this part. If 25% of ARA result still patent despite occlusion treatment this is highly relevant and has to be discussed further including comparison to existing literature and also mentioned in the conclusion.

As reported in the Method section: “Technical success was delineated by the successful implantation of a stent graft without the need for surgical conversion, intraoperative mortality, type I or III endoleaks, and any evidence of stent graft migration or occlusion right after the operation.” In our study it was 100%. ARAs was not embolized in all the cases presented. And the 25% of ARAs were left untreated, because it was thought that they do not need treatment.

Line 206: Follow-up should be reported either as mean +- standard deviation or as median and range. Further, with a mean FU of about 24 months you can only speak of mid-term but not long-term outcomes. Please also add information about completeness of FU / patient drop-outs. The follow-up information has been added.

Line 208-2014: You describe first ARA and lumbar artery involvement, then later also the IMA was also ligated. Was the IMA also contributing to the type 2 EL from the beginning? Or was there no retrograde perfusion from IMA to the aneurysm sac in the beginning? Yes the involvement was present since the beginning and it was added in the Manuscript-

Often, in such cases with aneurysm sac growth despite type 2 EL treatment, unrecognized type Ia or Ib endoleak was present from the beginning. In such cases, additional CEUS may help in the diagnostic process. What type of EVAR was implanted in this case, was it inside IFU regarding proximal neck characteristics? The EVAR implantation was performed with a Gore graft and the implant was performed inside the IFU

Line 217: Why was distal extension required in two patients? Type Ib endoleaks? Yes, for Type Ib endoleak.

In general: How many of your patients with type 2EL had communications between ARA and/or lumbar arteries and/or the IMA? This information was added to the text.

And have you got information on anticoagulation therapy in the patients of your study cohort? We do not have information about  the therapy.

Both aspects are important to note and discuss, since they may contribute to endoleak persistence requiring embolization. I suggest adding this to your analysis. It was added to the discussion section.

Discussion:

Lines 268-274: Here you discuss correctly the relevance of communicating arteries in type 2 EL persistence with respective references. However, the prevalence of multiple arteries in your study should be described more precisely, in the methods section (see commentaries above) and here. The prevalence of multiple arteries has been added to the Method section.

Line 297-299: I suggest rephrasing this sentence as “…was most likely a secondary outcome…” because you cannot be entirely sure that this was the case. Even if CT scans did not show contrast media in the aortic neck at the level of the ARA, may still have contributed to the type IA endoleak in terms of endotension. The sentences has been revised according to your suggestion.

Lines 300-305: Please discuss your results compared to those of Sadeghi-Azandaryani et al. Our results has been discussed compared to Sadeghi-Azandaryani.

Lines 309-312: How was the resolution of CT scans in your series? Were they all 1mm slices and were they all contrast-enhanced? This has to be clarified also in the methods section. The explanation has been added.

Line 305: “2” has to be removed Removed

Line 309: “17” has to be removed Removed

You should comment on kidney-protection perioperative strategies to prevent renal function deterioration in case of ARA covering/coilig. Please also comment the possibility of ARA preservation by FEVAR procedures versus the risk of occlusion of small stented ARAs in FEVAR. The comment on possible FEVAR procedures has been added.

Further, since you recommend in the conclusion a heightened post-EVAR monitoring regimen, you should expand on this in the discussion. How would it look like? Our protocol has been added to the conclusion section.

Discussion of study limitations is largely missing, please revise. The study limitation has been added.

Reviewer 2 Report

Comments and Suggestions for Authors

Informative review of a single institutional group.  I have several comments:

-when reporting percentages you should use a period rather than a comma (i.e. line  198 should be 33.3% rather than 33,3%)

-line 199 I think that aneurysm enlargement of 5 mm in 30 days from a type 11 endoleak is rather unusual.  Do you think this is correct or a problem with imaging

-line 191 why do think your length of stay was so long (8.78 days). I would expect that the LOS would be subsantially shorter.

-at this time do you routinely embolize all ARA >3 mm

Comments on the Quality of English Language

Minor corrections required

Author Response

Dear reviewer, 

The Manuscript has been revised according to your suggestion and answered every comment below. 

-when reporting percentages you should use a period rather than a comma (i.e. line  198 should be 33.3% rather than 33,3%) The text has been revised accordingly.

-line 199 I think that aneurysm enlargement of 5 mm in 30 days from a type 11 endoleak is rather unusual.  Do you think this is correct or a problem with imaging. This patient’s type II endoleak lead to type 1A endoleak, in further imaging, but at the six month CTA, it was not possible to detect the presence of a different kind of endoleak than type 2.

-line 191 why do think your length of stay was so long (8.78 days). I would expect that the LOS would be subsantially shorter. Our length of stay is influenced by long preoperative hospitalization for preoperative work-up.

-at this time do you routinely embolize all ARA >3 mm. Yes, we routinely performed ARAs embolization, in case of ARAs emerging from the aneurysmal sac.